# Ending the TB Crisis in Low- and Middle-Income Countries of the Eastern Mediterranean Region—Overcoming Inaction Through Strategical Leaps

**DOI:** 10.3390/tropicalmed10120348

**Published:** 2025-12-12

**Authors:** Santosha Kelamane, Ghada Muhjazi, Nevin Wilson, Martin van den Boom

**Affiliations:** Regional TB Programme Group, WHO Regional Office for the Eastern Mediterranean, Cairo 11371, Egypt; muhjazig@who.int (G.M.); wilsonn@who.int (N.W.); vandenboomm@who.int (M.v.d.B.)

**Keywords:** tuberculosis, domestic financing, people centered care, cross border collaboration

## Abstract

Tuberculosis (TB) remains a public health threat in low- and middle-income countries (LMICs) of the World Health Organization (WHO) Eastern Mediterranean Region (EMR), driven by a combination of social determinants including undernutrition, fragile health systems, conflict-related disruptions, human mobility and displacement, sub-optimal programmatic implementation, and insufficient domestic investment. These programmatic and governance constraints operate within a broader geopolitical context marked by conflict, sanctions, protracted crises, and large-scale displacement, which further limit countries’ ability to deliver uninterrupted TB services. In 2023, the region’s TB incidence was estimated at 116 per 100,000 population, with Pakistan alone accounting for about 73% of the regional burden. Despite a multitude of efforts, progress in reducing the TB burden in the EMR remains slow, with high case detection and treatment coverage gaps, low uptake of TB preventive treatment (TPT), underutilization of WHO-recommended rapid diagnostics, and only 25% of drug-resistant TB (DR-TB) cases initiated on treatment. Vulnerable populations, including internally displaced persons, migrants, refugees, prisoners, and returnees, continue to face major access barriers, and cross-border TB collaboration remains limited. This commentary reasons that the slow pace of TB burden reduction in the region is not only a biomedical or resource issue but also a reflection of structural and governance shortcomings. It proposes a ten-point strategic vision focused on building a sustainable ecosystem, enhancing primary healthcare systems, adopting people-centered and rights-based approaches, leveraging artificial intelligence, and gradually reducing dependency on external donors where feasible. However, in highly fragile settings such as Yemen or Somalia, domestic financing remains limited, and sustained external support will continue to be indispensable. The commentary calls for stronger national leadership, inclusive stakeholder engagement, and increased domestic financing to deliver integrated and resilient TB services. Ending TB in the EMR is within reach, but it requires boldly committed, coordinated, and country-led action.

## 1. Introduction

The tuberculosis action plan for the WHO Eastern Mediterranean Region 2023–2030 contains an ambitious vision to achieve the End TB goals, with approaches tailored to its heterogenous TB burden [1]. However, the epidemiological disparities, low treatment coverage, and suboptimal access to rapid diagnostics and persistent structural barriers are hindering TB service delivery to the patients. These include limited access to care not only for the general population but also for internally displaced persons, prisoners, returnees, and other vulnerable populations; low uptake of TPT due to weak screening and contact tracing mechanisms; and inadequate cross-border collaboration that hinders continuity of care for mobile populations. These access-related challenges further intensify inequities and hamper regional efforts to end TB.

## 2. Drug-Susceptible TB

As shown in Figure 1 below, in 2023, the estimated TB incidence rate in the EMR was 116 (95% UI: 89–145) per 100,000 population. Pakistan accounted for approximately 73% of the region’s TB burden, with an estimated incidence rate of 277 (188–368) per 100,000 population, followed by Somalia at 243 (151–355), Djibouti at 218 (range: 167–276), and Afghanistan at 180 (112–263) per 100,000 population. The estimated TB mortality rate in the region was 10 (8.8–12) per 100,000 population among HIV-negative individuals and 0.28 (0.22–0.34) among people living with HIV. The highest mortality rates among the HIV positive and negative individuals were observed in Somalia at 64 (37–99), Afghanistan at 24 (15–37), Djibouti at 25 (16–35), Pakistan at 20 (16–24), and Libya at 13 (7.7–20) per 100,000 population [2].

However, only 638,521 TB cases were notified out of 936,000 (95% UI: 723,000–1,200,000) incident TB cases in 2023, leaving 32% incident cases undiagnosed/not reported, with the potential to continue spreading the disease in the community and contributing to increased morbidity and mortality. The notification gap remains high in Somalia (58%), Sudan (47%), Libya (46%), Afghanistan (34%), and Pakistan (31%), which is concerning. Globally, the percentage of TB-affected households facing catastrophic costs are estimated to be 49% (38–60, pooled estimate 2014–2023), with the burden remaining high in Afghanistan (94%), Sudan (70%), Somalia (68%), and Pakistan (58%), indicating that the TB patients and their households suffer economically [2].

Despite commitments under the End TB Strategy and UNHLM TB targets, progress in the EMR lags behind global achievements and other WHO regions. While globally, TB incidence fell by 8.3%, the EMR achieved only a 3.4% reduction from the 2015 baseline to 2023. The reduction in incidence and death achieved in the EM region was lowest among all other regions—the highest reduction in TB incidence of 27% is seen in the WHO African Region, followed by a 27% reduction in WHO Europe. Similarly, the TB mortality declined by a meager 7% instead of the targeted 75% by 2025, with the highest reduction of 42% seen in the WHO African Region, followed by 38% reduction in WHO Europe. The scale-up of TPT is not on track to achieve the targets—coverage for household contacts and people living with HIV on ART started on TPT remains as low as 6.2% and 21%, respectively as shown in Figure 2 [2,3,4].

A major concern is the decline in the use of WHO-recommended rapid molecular diagnostics for new and relapse TB cases. In 2022, 55% of newly diagnosed patients received rapid testing; however, this dropped to 45 percent in 2023, marking a 10% decline that reverses years of progress. This regression has led to continued reliance on smear microscopy, which is significantly less susceptible and delays accurate diagnosis [2,5].

## 3. Drug Resistant TB

The programmatic management of drug-resistant TB has been adopted by all countries in the region and has shown notable progress over time. However, significant challenges persist. In 2023, treatment was initiated for only 5245 DR-TB cases, representing one quarter of the estimated 21,000 cases (95% UI: 16,000–27,000). Consequently, nearly three out of every four people with DR-TB did not receive treatment. Furthermore, only 24% of those treated received the latest all-oral shorter regimens, which are equally effective and less toxic than conventional longer treatments. Substantial gaps remain in both the diagnosis and treatment of DR-TB across most countries in the region [2,6].

## 4. Why Has TB Reduction Been Slow?

We deliberate on the reasons that are contributing to challenges in reducing the TB burden in EMR.

### 4.1. Sub-Optimal Domestic Investments in Public Health

Despite decades of global investment, TB reduction in many LMICs remains sluggish due to insufficient domestic financing. Many countries continue to rely heavily on international donors, with the Global Fund being the only major funding source in many settings. As the Global Fund’s investment priorities have sharpened and cost of business increased, lower priorities, which often include ‘soft’ interventions do not get funded, compromising TB prevention and control. Domestic contributions to TB programs remain disproportionately low, undermining sustainability and national ownership. This over-dependence on external technical and funding agencies has constrained countries from developing resilient, self-sufficient health systems that can adapt to evolving TB challenges [7].

The recent halt in U.S. humanitarian aid to WHO and other multilateral partners has further widened the funding gap, service delivery, and program continuity. However, this also presents an opportunity for governments to reassess their public health priorities and increase domestic investments to reduce reliance on donor-driven priorities.

### 4.2. Implementation Issues in Public Health

Effective TB control has been undermined by systemic and operational failures across multiple levels of the health system.

Leadership Gaps: There is a critical shortage of public health leadership at the district, provincial, and national levels. Inconsistent program stewardship often leads to poor prioritization, fragmented implementation, poor surveillance, limited knowledge of accurate disease burden, potential under diagnosis, and misalignment with evolving epidemiological needs [8,9].Failure to Prioritize Programmatic Actions: Countries frequently struggle to identify and act on high-impact interventions. TB strategies often focus on a narrow set of interventions, rather than adopting a holistic, patient-centered approach that addresses all points in the care cascade [8,9].Limited Community Involvement: Community and civil society engagement remains weak, despite their crucial role in case detection, adherence support, and reducing stigma. The exclusion of TB survivors, local networks, and community health workers is a missed opportunity to build people-centered care models [10].Infrastructure Constraints: This remains a significant barrier to effective TB control in many EMR LMICs. Limited access to molecular diagnostics results in delayed or missed diagnosis, particularly for drug-resistant TB, which hinders timely treatment. Capacity-building for health workers is often overlooked, leading to inconsistent clinical practices and suboptimal care delivery. Frequent stockouts and inefficient procurement systems disrupt the availability of essential TB medications and undermine treatment continuity. Patient support systems, including nutritional, psychosocial, and financial assistance, are insufficiently developed, leaving vulnerable populations without the support they need. Additionally, human resource challenges persist due to low remuneration, limited incentives, and heavy workloads, resulting in poor motivation and high attrition among health staff [11].

### 4.3. Lack of Operational Research—Linking Evidence to Policy

There is a significant gap between evidence and policy in TB prevention and control. Operational and implementation research is underfunded, and where it exists, its findings (evidence-generated) are rarely translated into actionable policies. This gap results in outdated program strategies that fail to reflect real-time challenges, local epidemiology, or emerging innovations such as AI-based diagnostics and digital adherence tools. Addressing this requires a deliberate push for investment in local research ecosystems, capacities, and mechanisms to embed evidence generation into national TB programs, ensuring policies are driven by context-specific data and innovations [8].

## 5. What Can Be Done to Change This Situation?

The authors propose ten bold shifts needed to accelerate reduction in TB burden and TB elimination in the region.

1.Create a sustainable multi-sector ecosystem:Mobilize national stakeholders including industries, manufacturers, policymakers, parliamentarians, academia, civil society, patient groups, and program implementers to establish a unified and comprehensive approach to TB elimination. The goal is to align national priorities, coordinate and direct resources, and ensure strong domestic accountability.2.Position the Ministry of Health as the anchor within the ‘multi-sector ecosystem’ with WHO support:Empower the Ministry of Health to lead TB response efforts, supported by WHO in providing evidence-based guidance, technical assistance and coordination, and facilitation of collaboration across the private sector, development partners, and government agencies through a multisectoral accountability framework [12].3.Bridge critical infrastructure gaps:Invest in strengthening primary healthcare systems, laboratory network optimization, integrated disease surveillance, and decentralized TB service delivery at primary healthcare levels including sample transportation from hard-to-reach areas. Special emphasis should be placed on expanding services to fragile and underserved areas in line with the principles of universal health coverage.4.Design fully people-centric strategies:Ensure that TB programs adopt a people-centered approach grounded in human rights. This includes addressing gender-related barriers, social stigma, poverty, displacement, and conflict while enhancing community engagement and psychosocial support. Special attention must be given to hard-to-reach populations such as refugees, migrants, IDPs, returnees, and prisoners, ensuring equitable access to diagnosis, treatment, and TPT [6].5.Empower communities through innovative delivery models:Engage civil society organizations and community health workers in TB prevention and care activities. Their involvement is essential for effective contact investigation, rollout of preventive treatment, and support for patient adherence and follow-up, particularly in hard-to-reach and conflict-affected areas. Strengthen mechanisms for cross-border TB collaboration to ensure uninterrupted care for mobile populations.6.Subsidize TB diagnostics and treatment in the private sector:Improve access to affordable TB services in the private healthcare sector by introducing targeted subsidies, implementing strategic purchasing mechanisms, and formally integrating private providers into national TB programs.7.Create sustainable financing mechanisms:Transition from heavy reliance on external donors by increasing domestic investments in health. This includes incorporating TB services into national health insurance schemes and engaging political leadership to advocate for sustained national funding.8.Return to programmatic basics:Prioritize the foundational elements of TB control, including early detection and effective treatment of drug-susceptible and drug-resistant TB. Strategies should focus on intensified case finding, systematic contact tracing, provision of TPT to contacts, HIV positive individuals, and other more-at-risk populations.9.Rapidly adopt artificial intelligence technologies to enhance TB screening:Leverage AI to improve early and accurate TB detection, especially in resource-limited and hard-to-reach settings. WHO-endorsed CAD software can be integrated with existing digital chest X-ray machines, enabling rapid, standardized, and high-throughput screening without relying on radiologists. Retrofitting current radiology infrastructure with CAD solutions offers a cost-effective approach to scaling AI-driven diagnostics across primary healthcare and mobile screening units [13].10.Ensure uninterrupted supply of drugs, consumables and reagents, and equipment maintenance to ensure quality diagnostics and treatment:Ensuring uninterrupted supply requires proactive forecasting, timely procurement, and decentralized stock monitoring. In addition, maintaining diagnostic and treatment equipment such as GeneXpert machines, digital X-ray units, and biosafety cabinets is critical to avoid service interruptions. Establishing national maintenance frameworks, vendor support agreements, and spare part stockpiles is essential to extend the functional lifespan of TB-related equipment. Integrating TB supply chains into broader essential health commodity platforms can improve efficiency, accountability, and responsiveness.

## 6. Conclusions

Moving from vision to action requires catalyzing TB burden reduction and accelerating elimination through national leadership and primary healthcare-integrated approach.

The path to ending TB in the Eastern Mediterranean Region’s low- and middle-income countries demands more than incremental programmatic improvements. It requires a 180-degree shift in the way TB is viewed, funded, and delivered, so that the TB Regional Action Plan 2023–2030 can demonstrate its full potential for impact. The TB crisis is not a biomedical challenge; it is a test of national governance, accountability, and multisectoral primary healthcare approach.

The current disruption in external donor support, including the stopping of USAID, provides opportunity and underscores the urgent need to transition toward sustainable, domestically driven TB responses. Rather than retreat, this moment should be seized to recalibrate national priorities, embed TB into broader public health and universal health coverage strategies, and invest in resilient systems that serve all.

To realize this vision, governments must take the lead in building a sustainable multi-sectoral ecosystem. This includes anchoring multisectoral coordination in ministries of health, supported by WHO and other partners, and engaging all relevant stakeholders in a unified effort, from industry and academia to civil society and cured patients.

A future-ready TB response must be rooted in primary healthcare, digitally enabled, patient-centered, and equity-driven. Investments in diagnostics, infrastructure, health workforce, private sector engagements, supply chain management, and community-led models are essential to bridge persistent gaps, especially in fragile and underserved settings.

The time for fragmented, donor-dependent, and siloed approaches is over. Ending TB in the region is possible only if countries commit boldly, invest sustainably, and deliver inclusively, in line with the 2025 World TB Day call to action: Yes, we can END TB—Commit, Invest, Deliver.

## Figures and Tables

**Figure 1 tropicalmed-10-00348-f001:**
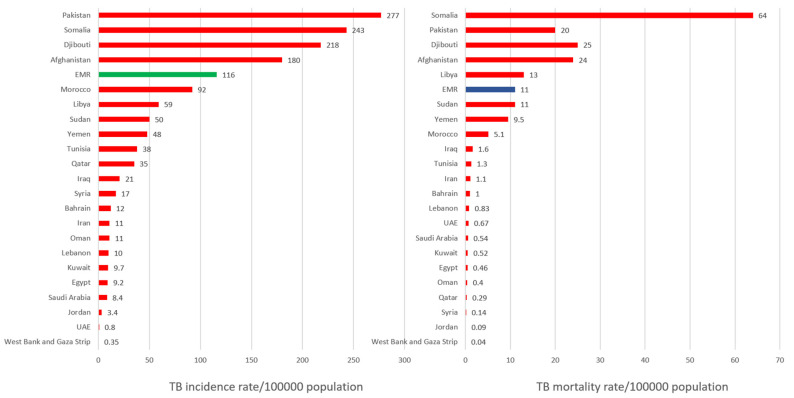
Estimated TB incidence and TB mortality in EMR in 2023.

**Figure 2 tropicalmed-10-00348-f002:**
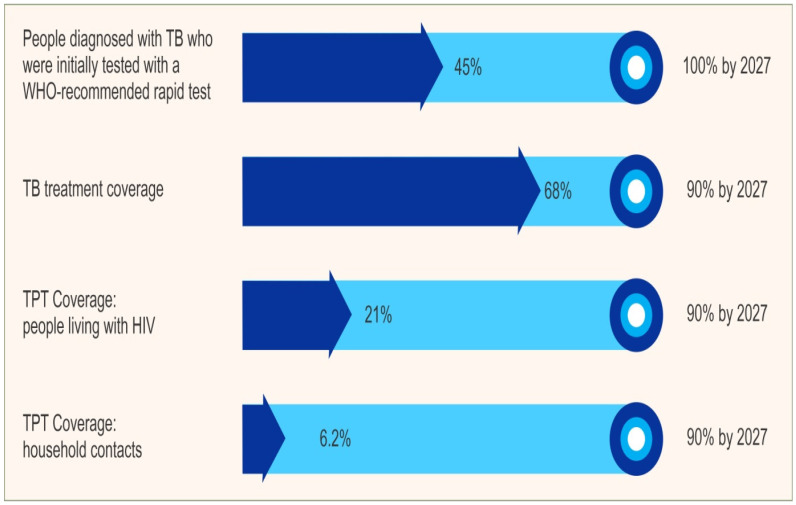
Progress against the UNHLM indicators (2023).

## Data Availability

No new data were created or analyzed in this study.

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
