# Peer review of "Ending the TB Crisis in Low- and Middle-Income Countries of the Eastern Mediterranean Region—Overcoming Inaction Through Strategical Leaps"

_tropicalmed, 2025, doi:10.3390/tropicalmed10120348_

Round 1
Reviewer 1 Report
Comments and Suggestions for Authors
Dear Authors,
I have reviewed your article titled “Ending the TB crisis in Low and Middle-Income countries of the Eastern Mediterranean Region – Overcoming inaction through strategical leaps”. Thank you for your efforts.
The study is a comprehensive commentary analyzing why progress in combating TB has been slow in the Eastern Mediterranean Region. It provides strong regional context, and data is drawn from recent WHO reports. The authors offer actionable policy recommendations. The article can serve as a valuable framework for public health policymakers.
Strengths
- Based on region-specific epidemiological data
- Offers structural critiques, particularly focusing on governance, financing, and health systems
- Well-organized and clear ten-point strategic recommendation section
- Good flow and readability
It is generally well designed and written. However, the following minör revisions should be made before it is accepted:
- Abbreviations should be explained at their first mention.
- Figures should be cited.
Best regards,
Author Response
Comment 1: It is generally well designed and written. However, the following minör revisions should be made before it is accepted.
Response 1: Dear Reviewer, thank you very much for reviewing our manuscript, your inputs are well taken and addressed accordingly.
Comment 2: Abbreviations should be explained at their first mention and Figures should be cited.
Response 2: Thank you, reviewer. This has been addressed, Thanks again

Reviewer 2 Report
Comments and Suggestions for Authors
- A commentary should go beyond reiteration and provide interpretation or "provocative" reflection. The “ten strategic shifts” are all conceptually valid, but they read as a restatement of WHO guidance rather than as a contextualized argument grounded in new analysis or field experience
- Statements such as “Ending TB in the EMR is achievable” may be overly optimistic and not reflect the heterogeneity in health systems, conflict intensity, and data quality.
- Figures 1 and 2 are mentioned but not methodologically described. The sources, year (2023), and data reliability (modeled vs. notified cases) are not explained
- There is a risk of “policy bias,” where governance or financing is blamed without acknowledging broader geopolitical drivers (war, displacement, sanctions)
- The commentary implicitly portrays the cessation of external donor support (e.g., USAID) as an “opportunity” for national ownership, but this could be misleading. In fragile states as Yemen or Somalia domestic financing is not feasible short-term.
- In the introduction, "...and it’s heterogenous TB burden” should be corrected to “its heterogeneous TB burden.”
- In "conflicts of interest", please correct "he authors"
- Use uniform format for references
- Even for a commentary, include a brief sentence under “Ethical considerations” clarifying that no human data were collected, to meet journal transparency norms
Author Response
Comment 1: A commentary should go beyond reiteration and provide interpretation or "provocative" reflection. The “ten strategic shifts” are all conceptually valid, but they read as a restatement of WHO guidance rather than as a contextualized argument grounded in new analysis or field experience.
Response 1: Thank you reviewer for reviewing our manuscript. Although many of the proposed shifts align with existing WHO guidance, their significance in the EMR context lies in how they respond to region-specific barriers like protracted crises, political instability, chronic underfunding, and fragmented governance. The insights presented here draw from field missions, programmatic reviews, regional monitoring data, and operational challenges observed across these countries. However we considered your inputs and revised the manuscript.
Comment 2: Statements such as “Ending TB in the EMR is achievable” may be overly optimistic and not reflect the heterogeneity in health systems, conflict intensity, and data quality.
Response 2: Thank you for your comment. This has been addressed and changed to "within the reach"
Comment 3: Figures 1 and 2 are mentioned but not methodologically described. The sources, year (2023), and data reliability (modeled vs. notified cases) are not explained
Response 3: Thank you for this important observation. We have now added a methodological description for Figures 1 and 2, specifying the data sources, the reference year (2023), and clarification on whether estimates are modeled or based on notified cases.
Comment 4: There is a risk of “policy bias,” where governance or financing is blamed without acknowledging broader geopolitical drivers (war, displacement, sanctions)
Response 4: Thank you for very important inputs. This is been addressed and included in line no 13 to 16.
Comment 5: The commentary implicitly portrays the cessation of external donor support (e.g., USAID) as an “opportunity” for national ownership, but this could be misleading. In fragile states as Yemen or Somalia domestic financing is not feasible short-term.
Response 5: Thank you again. This has been addressed and included in line no 29 to 32.
Comment 6: In the introduction, "...and it’s heterogenous TB burden” should be corrected to “its heterogeneous TB burden.”
Response 6: Thank you and this has been addressed in line no 40 to 41.
Comment 7: In "conflicts of interest", please correct "he authors"
Response 7: Thank you. This is addressed in line no 250
Comment 8: Use uniform format for references
Response 8: Thank you. All the references are reformatted uniformly.
Comment 9: Even for a commentary, include a brief sentence under “Ethical considerations” clarifying that no human data were collected, to meet journal transparency norms
Response 9: Thank you. "Ethical considerations – No human data were collected" is added in line no 251.

Reviewer 3 Report
Comments and Suggestions for Authors
A very insightful and well-structured contribution with strong recommendations, especially as many of the TB high burden countries in conflicts or active security challenges.
However, it’s important to highlight that most of the low-performing countries in the region share a common underlying factor—fragility and ongoing conflict or security challenges—which appears to have received limited emphasis in the commentary.
Active conflict profoundly disrupts service delivery, constraining the movement of patients, healthcare staff, and essential supplies. It also undermines facility-based diagnostic networks, limits access to care, and complicates cross-border TB management. Moreover, the proliferation of humanitarian actors and parallel service structures, often with weak integration of TB services, further fragments care and accountability.
Addressing these contextual realities is critical for understanding performance gaps and for designing flexible, community-linked, and integrated approaches tailored to fragile and conflict-affected settings.
Author Response
Comment from the reviewer: A very insightful and well-structured contribution with strong recommendations, especially as many of the TB high burden countries in conflicts or active security challenges. However, it’s important to highlight that most of the low-performing countries in the region share a common underlying factor—fragility and ongoing conflict or security challenges—which appears to have received limited emphasis in the commentary. Active conflict profoundly disrupts service delivery, constraining the movement of patients, healthcare staff, and essential supplies. It also undermines facility-based diagnostic networks, limits access to care, and complicates cross-border TB management. Moreover, the proliferation of humanitarian actors and parallel service structures, often with weak integration of TB services, further fragments care and accountability. Addressing these contextual realities is critical for understanding performance gaps and for designing flexible, community-linked, and integrated approaches tailored to fragile and conflict-affected settings.
Response: We appreciate this valuable observation. We have now strengthened the commentary by explicitly highlighting fragility, conflict, and security challenges as key determinants of TB program performance in the EMR.

Reviewer 4 Report
Comments and Suggestions for Authors
The work presented addresses a current problem from a perspective that is not always considered, which gives it great value. These are my observations that I believe can improve the manuscript:
1. Figures 1 and 2 should be cited within the text.
2. In line 59, remove (68%); this data is unnecessary since the percentage of unattended cases is mentioned within the same sentence.
3. Do the percentages in line 63 refer to unreported cases? Please clarify.
4. Regarding the idea presented in lines 79 to 82, what do the authors believe could explain the rejection of molecular tests? Or is it merely due to a lack of access to them?
5. In the sentence in lines 88 to 90, "In 2023...", the idea is not very clear; its wording needs improvement.
6. The 10 suggestions presented in the section "What can be done..." can be presented in a table, where one of the columns lists the issues presented in the previous section, "Issues in public health," thus showing which problems can be resolved with the proposed ideas. The table should also be cited in the text.
Author Response
Comment 1. Figures 1 and 2 should be cited within the text.
Response 1: Thank you for reviewing our manuscript. This is very important observation and it has been addressed.
Comment 2. In line 59, remove (68%); this data is unnecessary since the percentage of unattended cases is mentioned within the same sentence.
Response 2: Thanks again, This has been addressed (Line no 64)
Comment 3. Do the percentages in line 63 refer to unreported cases? Please clarify.
Response 3: Thank you. Yes, this is unreported cases.
Comment 4. Regarding the idea presented in lines 79 to 82, what do the authors believe could explain the rejection of molecular tests? Or is it merely due to a lack of access to them?
Response 4: Thank you for this observation. The decrease can be attributed to health systems barriers to access funds from the National Government or donor agencies.
Comment 5. In the sentence in lines 88 to 90, "In 2023...", the idea is not very clear; its wording needs improvement.
Response 5: Thank you. This has been addressed. the revised text is "In 2023, treatment was initiated for only 5,245 DR-TB cases, representing one quarter of the estimated 21,000 cases (95% UI: 16,000–27,000). Consequently, nearly three out of every four people with DR-TB did not receive treatment". Line no 93 to 95.
Comment 6. The 10 suggestions presented in the section "What can be done..." can be presented in a table, where one of the columns lists the issues presented in the previous section, "Issues in public health," thus showing which problems can be resolved with the proposed ideas. The table should also be cited in the text.
Response 6: Thank you for your suggestions and we appreciate it. However, we feel that presenting it as a table will take away the essence of audience eye-balling the narrative. Hence, we wish to keep it the way it is and we strongly believe that you too will agree on this.

Round 2
Reviewer 2 Report
Comments and Suggestions for Authors
Thanks to the authors for their work, I think this manuscript reflects more the stagnant situation physicians are facing in TB control.
Comments on the Quality of English LanguageNo further comments.